# Half Squat Mechanical Analysis Based on PBT Framework

**DOI:** 10.3390/bioengineering12060603

**Published:** 2025-06-01

**Authors:** Miguel Rodal, Emilio Manuel Arrayales-Millán, Mirvana Elizabeth Gonzalez-Macías, Jorge Pérez-Gómez, Kostas Gianikellis

**Affiliations:** 1BioẼrgon Research Group, University of Extremadura, 10003 Cáceres, Spain; kgiannik@unex.es; 2Laboratory Biomechanics, Faculty of Sports, Autonomous University of Baja California, Mexicali 21289, BC, Mexico; earrayales@uabc.edu.mx (E.M.A.-M.); gonzalez.mirvana@uabc.edu.mx (M.E.G.-M.); 3Health, Economy, Motricity and Education (HEME) Research Group, University of Extremadura, 10003 Cáceres, Spain; jorgepg100@unex.es

**Keywords:** half squat, power-based training, mechanical power, exercise biomechanics

## Abstract

Muscular strength is an essential factor in sports performance and general health, especially for optimizing mechanical power, as well as for injury prevention. The present study biomechanically characterized the half squat (HS) using a systemic structural approach based on mechanical power, called Power-Based Training (PBT), through which four phases of the movement were determined (acceleration and deceleration of lowering and lifting). Five weightlifters from the Mexican national team (categories U17, U20, and U23) participated, who performed five repetitions per set of HS with progressive loads (20%, 35%, 50%, 65%, and 80% of the one repetition maximum). The behavior of the center of mass of the subject–bar system was recorded by photogrammetry, calculating position, velocity, acceleration, mechanical power, and mechanical work. The results showed a significant reduction in velocity, acceleration, and mechanical power as the load increases, as well as variations in the duration and range of displacement per phase. These findings highlight the importance of a detailed analysis to understand the neuromuscular demands of HS and to optimize its application. The PBT approach and global center of mass analysis provide a more accurate view of the mechanics of this exercise, facilitating its application in future research, as well as in performance planning and monitoring.

## 1. Introduction

Muscular strength plays a fundamental role both in the field of high-performance sport and in the promotion of general health throughout life. In the sportive context, its importance is particularly notable in elite athletes, as it enables effective distribution of force across muscle chains, a key element in disciplines such as football and athletics [1,2]. Furthermore, it has been identified as a reliable predictor of athletic performance, which has led to its systematic inclusion in training programs designed not only to optimize physical performance, but also to prevent injuries [1].

Beyond competitive sport, muscular strength is a significant biological indicator of general health from early in life. It is associated with other components of physical fitness, such as speed, agility, and aerobic capacity, thus establishing a solid physiological basis for balanced physical development [3]. In older adulthood, its maintenance becomes critically important as it is essential for the performance of daily activities and the prevention of debilitating conditions such as sarcopenia, characterized by the progressive loss of muscle mass and function [4,5].

Moreover, previous studies have shown additional benefits of muscle strength in the cognitive and behavioral domains. In children, particularly in children diagnosed with attention deficit hyperactivity disorder (ADHD), greater muscle strength has been found to correlate positively with better cognitive performance and more adaptive behavior in school settings [6].

Consequently, strength training has become a constant presence in a variety of contexts, from sports development programs to public health interventions. Among the commonly employed methods, weight training has proven to be an effective strategy for improving both physical fitness and athletic performance [7] and is commonly used by athletes to enhance maximum strength and muscle power [8]. In this context, scientific literature broadly supports weight training as a versatile tool, capable of inducing key physiological adaptations in multiple dimensions of neuromuscular performance. Its systematic practice has been shown to be effective in improving maximal strength [9,10], muscle power [8,11], hypertrophy development [9,10], and fatigue resistance [12,13]. These benefits are not only manifested in the sporting community but have also been exploited in clinical and therapeutic settings, where weight training has shown positive effects both in functional rehabilitation processes [14,15,16,17,18] and in post-injury sports rehabilitation.

In particular, it has been identified that strength training performed at high speed produces superior improvements in functional performance and in the expression of muscle power, compared to methods based on low execution speeds [4]. This finding emphasizes the need to consider not only the magnitude of the load, but also the execution speed as a determining variable in the planning of strength training, especially when the objective is to optimize performance in explosive or functionally demanding motor actions.

Within this framework of strength training, the squat has become one of the most widely used exercises for the development of the lower body, given its effectiveness in improving strength, muscle hypertrophy, and maximum strength in the lower limbs [19,20]. Its versatility has allowed its inclusion in both conventional training programs and in more advanced proposals that integrate different load intensities [21]. Among its many variants, the front squat and the back squat have shown comparable effects in terms of increasing maximal dynamic strength and muscle mass development [22]. Likewise, the half squat (HS) represents a particularly useful option in contexts where the depth of movement needs to be limited, either due to specific training goal considerations or individual mobility restrictions [23,24]. The involvement of large muscle groups, such as glutes and quadriceps femoris, reinforces its value in both increasing athletic performance and injury prevention [25]. This quality makes the squat an indispensable tool for strength training [26,27,28,29]. However, due to the technical complexity of its execution, rigorous control of the biomechanical variables determining its effectiveness is necessary to maximize the benefits and minimize the risk of injury.

According to the technical description proposed by Pérez-Castilla [30], the HS consists of a controlled descent until reaching a 90° angle at the knee joint, ensuring full contact of the feet with the ground. This is followed by an ascent to an upright position, both phases being performed at the maximum possible voluntary velocity.

Nowadays, this exercise is frequently analyzed under the velocity-based training (VBT) approach, a paradigm that has become increasingly prominent in academic and professional practice [31,32]. Under this approach, the movement is segmented into two phases: an eccentric or ‘lowering’ phase and a concentric or ‘lifting’ phase [33,34,35,36]. Within this framework, parameters such as average and maximal velocity [37,38], average and maximal mechanical power [37], velocity loss [39,40], and force–velocity profile [41] are assessed.

However, for a correct treatment of the concept of ‘strength’, it is essential to clearly define strength and its qualities: muscular strength is defined as the capacity of the neuromuscular system to generate tension in order to overcome, resist, or counteract an external load by means of active muscular contractions [42]. Its level of development is a direct determinant of physical performance, which is why its monitoring, assessment, and improvement are essential within the field of sports training. Traditionally, the assessment of muscular strength has been structured around three main indicators, also known as qualities of strength: fatigability, maximal strength, and mechanical power [42].

Considering the intensity of the exercise (carried load) as an indicator of performance, it is important to distinguish between the three qualities of muscular force [43]: strength, power [44] and fatigability [42,45], as well as between mechanical work (W) and mechanical power (P) [46,47].

Given that this study does not include the evaluation of fatigability, which requires prolonged or repetitive efforts over time, or maximum strength, which involves the execution of submaximal or maximal isometric contractions, it will focus exclusively on the third quality: mechanical power. This variable, together with other relevant biomechanical parameters, constitutes the focus of the analysis carried out in this research.

In physical terms, and in relation to the center of mass (CoM) of a body segment or a multi-segment system, mechanical power (MP) is defined as the ratio between the mechanical work undertaken and the time taken to do it. This ratio can be expressed, equivalently, as the product of the applied force (measured in newtons) and the velocity of the displacement (measured in meters per second) [47]. In real human movement contexts, where force rarely remains constant over time, both force magnitude and execution speed vary considerably over time [47]. Consequently, to obtain an accurate estimate of the mechanical power developed, it is necessary to treat it as a function of time (Equation (1)). From this function, the calculation of the total mechanical work performed is obtained by integrating the area under the mechanical power curve over time (Equation (2)), which makes it possible to quantify the total mechanical cost of the movement analyzed [47].

Thus, the mechanical power is calculated as follows:(1)$$P=\frac{dW}{dt}=\vec{F}\cdot \;\frac{d\vec{s}}{dt}=\vec{F}\cdot \vec{v}=m\vec{a}\cdot \vec{v},$$
where *P* is the instantaneous mechanical power, expressed in watts; $\vec{F}$ is force, in Newtons; $\vec{v}$ is the velocity, in meters per second, and $\vec{a}$ is the acceleration, in meters per second squared.

The mechanical work is also calculated as:(2)$$W=\int_{t_{1}}^{t_{2}}P\;dt$$
where *W* is the mechanical work, expressed in joules (J); and *P* is the mechanical power, in watts.

Having determined the necessary theoretical foundation, this article proposes the study of the HS from the perspective of biomechanical analysis of muscle strength qualities. Despite its relevance, HS has been less studied in comparison with other variants of the squat, which justifies the need for its biomechanical parameterization. To carry out this characterization, it is necessary to determine a methodological procedure that allows the description of the biomechanical behavior in the HS, exploring the main phases of the exercise.

To this end, the Power-Based Training (PBT) framework is proposed. PBT is a methodological approach that allows the assessment and training of muscular strength qualities [42], based on the analysis of mechanical power [47] and the use of structural systemic analysis [48]. In the present work, the PBT approach is applied to the assessment of the mechanical power of the end-effector, i.e., the CoM of the subject–bar system. Sign changes in mechanical power indicate shifts in movement dynamics that allow a detailed analysis of the phases of the movement that occur during its execution. This methodology is particularly applicable in countermovement exercises, which integrate a stretch-shortening cycle consisting of an immediate transition between the eccentric and concentric phases [49]. This segmented analysis facilitates the quantification of relevant parameters for the understanding of physical effort, such as average and maximum mechanical power, as well as mechanical work, in each repetition, resulting in positive (concentric) and negative (eccentric) values depending on the phase of the movement.

In this context, the present study aims to (1) establish a systemic structural analysis [48] procedure for the HS exercise, in order to biomechanically determine its movement phases and its kinematic and kinetic parameters; and (2) evaluate them under different loading conditions. The evaluated parameters include the (vertical component of) CoM range of motion, velocity, acceleration, mechanical power, and mechanical work.

It is hoped that the results of this research will contribute to a better understanding of the factors that influence the execution of the HS exercise—and other similar lower body exercises with countermovement—providing detailed information on its mechanics in each phase of the movement.

## 2. Materials and Methods

### 2.1. Experimental Design

A cross-sectional study was conducted with the aim of biomechanically characterizing the motor pattern of the Half Squat (HS) exercise.

Given that velocity and acceleration can be affected by the intensity [46,50], it is essential to conduct a comparison and analysis across different load intensities during the HS exercise. For this purpose, an incremental loading protocol based on the known value of one repetition maximum (1RM) was used for each subject. 1RM is defined as the maximum amount of weight a person can lift for one complete repetition, executed with proper technique [51]. It started with a load equivalent to 20% of the 1RM and ended at 80%, with progressive increments of 15% (20%, 35%, 50%, 65%, and 80% of the 1RM).

The most recent known 1RM value of the full squat was selected as a reference, based on the established linear relationship between different squats [52]. This choice helped to avoid potential alterations in movement patterns that could arise from the higher technical demands of the full squat compared to the half squat. Additionally, it prevented an excessive fatigue accumulation throughout the protocol, given the greater mechanical effort required by the full squat. Since the focus of this study is to analyze the movement phases of a prototypical lower-body exercise, such as the HS, and the variation in biomechanical parameters across different load levels, the final range of motion was omitted, acknowledging the potential bias this may introduce in the magnitude of the parameters evaluated.

During the testing procedure, kinematic and dynamic data were recorded for five repetitions of HS for each load level (%1RM), in order to calculate the mean and maximum values of the following biomechanical variables (normalized according to the subject’s body weight), in each of the identified movement phases: (1) Vertical position of the CoM; (2) Vertical velocity of the CoM; (3) Vertical acceleration of the CoM; (4) Mechanical power of the CoM; (5) Mechanical work of the CoM. First and last repetitions were omitted from the analysis to avoid alterations due to the beginning and end of the sequence.

### 2.2. Subjects

With the aim of guaranteeing technical homogeneity and biomechanical validity in the execution of the HS exercise, a purposive sample was selected consisting of five professional athletes belonging to the Mexican national weightlifting team. All of them were competitors in bodyweight categories of 96 kg and 102 kg, within the age ranges of the U17, U20, and U23 divisions.

The inclusion criteria were the following: (1) belong to the national weightlifting team in the U17, U20, or U23 categories; (2) have a ratio between the maximum repetition value (1RM) in full squat and body weight greater than 1.5; (3) perform regular squat exercise training at least three times a week; (4) have no lower body or trunk injuries during the last year; and (5) have no medical contraindications for performing maximal efforts.

Subjects had an average body weight of 96.09 ± 4.34 kg, an average height of 172.88 ± 6.43 cm, an average full squat 1RM of 220.20 ± 28.73 kg, and a mean 1RM to body weight ratio of 2.29 ± 0.23.

The study was approved by the Department of Teaching and Research Support of the Mexicali Campus of the Autonomous University of Baja California, in accordance with the ethical principles established in the Declaration of Helsinki. All participants were duly informed about the study procedures and signed an informed consent form prior to participation.

### 2.3. Instrumental

For data collection in the present study, the Vicon biomechanical analysis system [53] and version 2.16 of Nexus software [54] were used, a reference tool in the three-dimensional capture of human movement, widely validated in both clinical and biomechanical research contexts. This system integrates high-speed optical technology with advanced reconstruction algorithms, enabling accurate and detailed assessment of kinematic and kinetic variables, with high spatial and temporal resolution.

The VICON system operates using a set of high-speed infrared cameras that detect the position of reflective markers strategically placed on the subject’s body surface. From the signals captured by these cameras, a three-dimensional reconstruction of the movement is generated based on the spatial trajectories of the markers, allowing the motor patterns involved in the execution of the exercise to be analyzed with great precision.

In the present study, the Plug-in Gait (PiG) Full Body [55] model was used, which allowed for the estimation of the vertical position of the CoM of the subject, as well as the vertical position of the CoM of the bar, determined through the average vertical position of the hands. From these data, it was possible to calculate the vertical position of the CoM of the complete system (subject + bar), a fundamental variable for the subsequent dynamic analysis.

### 2.4. Procedure

#### 2.4.1. Protocol

The biomechanical evaluation protocol of the HS exercise was structured in several consecutive phases. First, the participants’ personal data were collected, and the informed consent was signed, in accordance with the established ethical principles. Next, anthropometric measurements were recorded, and the loads for each set were calculated based on the 1RM value previously known for each athlete for the full squat exercise.

Subjects then underwent a general warm-up, consisting of low-intensity aerobic exercises and joint mobility, to prepare the body for the specific task. After this initial phase, the anatomical markers necessary for the photogrammetric recording were placed, following the anthropometric model mentioned in the section on instruments.

Once the preparation was completed, a static capture was performed with no external load, used as a biomechanical baseline to enable automated tracking of the markers during the dynamic analysis. This phase was followed by a specific warm-up that included progressive repetitions of the exercise under study with load levels below 20% of their 1RM.

Finally, the experimental protocol was implemented, consisting of the execution of the HS exercise under a progression of loads equivalent to 20%, 35%, 50%, 65%, and 80% of the 1RM. At each load level, participants performed five repetitions, with controlled rest periods of between 2 and 3 min between sets, in accordance with methodological recommendations to ensure complete recovery and avoid interference with performance [56,57,58].

#### 2.4.2. Signal Processing

Before proceeding to the calculation of the position of the CoM and its kinematic derivatives (velocity, acceleration, mechanical power, and mechanical work), a filtering process was applied to the original signal corresponding to the position of the reflective markers. A fourth-order Butterworth digital low-pass filter was used, with a cut-off frequency of 6.5 Hz [47,59,60].

#### 2.4.3. Determination of the Phases

Based on the PBT framework, this study hypothesizes that biomechanical analysis of the squat, applied to CoM mechanics, will allow an accurate segmentation of the movement into four phases, providing a detailed understanding of the dynamic demands of the exercise. Mechanical power, defined as the product of resultant force and velocity of the movement, is used as the central variable to differentiate between concentric actions, where mechanical power is positive, and eccentric actions, where mechanical power is negative. This approach combines vertical position, vertical velocity, and vertical acceleration of the CoM to identify key events in the movement cycle, considering positive values as upward-directed and negative values as downward-directed. The local minima and maxima of the vertical position of the CoM determine the moments of change in direction of the movement (when the velocity changes sign), just as the local minima and maxima of the vertical velocity of the CoM determine the moments of change in direction of the acceleration (when the resultant force changes sign) indicating whether the force exerted on the bar is greater or lower than the weight force of the bar, since the acceleration, associated with the resultant force by Newton’s second law, indicates the direction of the applied force. Finally, to determine the sign of the mechanical power, velocity and acceleration are combined as shown in Equation (1).

The four expected phases of the vertical movement are as follows:The first phase, lowering acceleration, is characterized by negative velocity and acceleration, where the resulting force acts in the same direction as the downward motion, generating positive mechanical power.The second phase, lowering deceleration, is defined by a negative velocity and a positive acceleration, at which point the resulting force opposes the downward movement, progressively reducing the velocity and producing negative mechanical power.The third phase, lifting acceleration, is distinguished by positive velocity and acceleration, where the resulting force cooperates with the upward movement, generating positive mechanical power and reflecting concentric mechanical work to overcome resistance.The fourth phase, lifting deceleration, presents a positive velocity and a negative acceleration, where the resulting force opposes the upward motion, progressively decreasing the velocity and the negative mechanical power as the cycle is completed.

The identification of these phases is based on the detection of the points where the CoM velocity changes sign, which delimit the transitions between descent and ascent, and on the points where the acceleration changes sign, which subdivide each half of the movement into acceleration and deceleration.

To ensure that the results could be compared with other studies with less precise methodologies in the division of the phases, the ‘rising’ phase was also considered as the set of phases 3 and 4.

### 2.5. Statistical Analysis

The mean and maximum values (understood as those with the greatest absolute magnitude) of the different biomechanical variables were calculated from the individual values obtained by each subject in each set (load condition), in each phase of the movement and in each repetition. To avoid possible alterations in the results derived from transitory variations at the beginning or end of the exercise, the first and last of the five repetitions performed in each set were excluded, considering only the three central repetitions. The aim of this methodological decision was to reduce the possible contamination of the results caused by the differences that these repetitions usually present with respect to others.

As for the inferential analysis, the non-parametric Friedman test was used to compare, within each biomechanical parameter, the values obtained between the different phases in each loading condition. This test allowed the detection of possible global differences between phases in dependent sample conditions.

In cases where the Friedman test yielded statistically significant results (*p* < 0.05), post hoc pairwise comparisons were conducted using the Wilcoxon test to identify the specific phases exhibiting significant differences. This sequence of analyses allowed a robust assessment of the evolution of the biomechanical parameters of the CoM throughout the phases of the movement, considering the non-parametric nature of the data and the small sample size.

## 3. Results

### 3.1. Phases of the Movement

Figure 1 illustrates the description of the squat exercise movement and its different phases, based on the mechanical power of the CoM of the subject with the barbell. According to the mechanical power results, the squat movement can be divided into four distinct phases. During phase 1 (P1), negative acceleration and velocity of the bar are observed, which translates into positive (concentric) mechanical power. However, it has been considered that the downward displacement of the CoM is mainly due to the action of gravity and not to the concentric action of the hip, knee, and ankle flexors. Phase 2 (P2) is characterized by positive vertical acceleration combined with negative vertical velocity, indicating eccentric mechanical power and consequently eccentric mechanical work of the hip, knee, and ankle extensors to slow the descent. During phase 3 (P3), positive acceleration is seen together with positive velocity, reflecting concentric action of the hip, knee, and ankle extensors, generating positive mechanical power to raise the CoM. Finally, phase 4 (P4) is characterized by negative acceleration with positive velocity, resulting in negative mechanical power, which could indicate eccentric action by the hip flexors and knee flexors; however, it has been observed that the braking behavior of the bar in this phase is mainly due to the action of gravity, although muscular activity may contribute partially to the control of the action, regulating the final range of ascent.

To enrich the understanding of the variation as a function of load, the movement analyzed is decomposed into 4 graphs in Figure 2, including the mean time series of each variable at each load condition, for one subject. These results show the generalized reduction in the magnitude of the variables analyzed as the load level increases.

Moreover, Table 1 presents a detailed analysis of the kinematic and dynamic variables associated with the execution of the HS (CoM behavior), showing the mean and maximum values of the variables studied. In this context, the maximum values correspond to those of greater absolute magnitude, regardless of their sign, as this depends on the direction of movement in each phase of the exercise.

The analysis of the squat movement has been structured in four phases, as shown in the preceding figure. The first phase corresponds to the acceleration of the descent, where the downward displacement of the bar begins. The second phase comprises the braking of the descent, in which the bar slows down until it reaches the transition point to the upward movement. The third phase is characterized by the acceleration of the upward movement, in which an increase in the positive velocity of the movement is generated. Finally, the fourth phase represents the braking of the upward movement, at which point the velocity of the bar decreases until the movement comes to a complete stop.

The results obtained for each phase and set have been organized in the table in two sub-columns. The first of these presents the mean values of each variable, with the significant differences identified to the right. The second sub-column shows the absolute maximum values recorded, which are also accompanied by the significant differences detected.

As for the variables analyzed, the average duration of each phase is expressed both in seconds and as a percentage of the total squat time. The position-related variable, defined as the range of displacement, represents the distance travelled by the CoM during each phase. The velocity is recorded with a positive sign when the displacement is vertically upward and with a negative sign when the displacement is vertically downward. Similarly, acceleration has positive values when the acceleration vector is directed upwards and negative values when the acceleration is directed downwards. For mechanical power and mechanical work, positive values reflect concentric action, while negative values correspond to eccentric action.

### 3.2. Time Parameters

Analysis of the duration of each phase of the HS cycle reveals a pattern of progressive change as the load increases. In the initial lowering phase (Phase 1), corresponding to the downward acceleration phase, the average time decreases noticeably from 0.42 s (28.9% of the cycle) at the lightest load (S1) to 0.31 s (21.4%) at the heaviest load (S5). Complementarily, the lowering deceleration phase (Phase 2) shows a slightly increasing trend from 0.34 s (25.0%) in S1 to 0.41 s (28.5%) in S5.

In the upward part, the positive acceleration phase (Phase 3) progressively lengthens from 0.28 s (22.0%) in S1 to 0.45 s (31.2%) in S5. In contrast, the final phase of the ascent (Phase 4), corresponding to upward braking, reduces its duration from 0.34 s (24.1%) in S1 to 0.27 s (18.8%) in S5. Thus, the total rise time (P3 + P4) increases from 0.63 s (46.1% of the cycle) in S1 to 0.72 s (50.0%) in S5, which emphasizes the slowing down of the ascent under higher mechanical demands.

### 3.3. Kinematics

The kinematic behavior of the CoM is evaluated by means of position, velocity, and acceleration, observing changes in each phase with the progressive increase in load.

In terms of *position*, the range of vertical displacement in the descent (Phases 1 and 2) does not show statistically significant differences, although in the initial phase (Phase 1), a slight downward trend is observed, with values between 0.17 m in S1 and 0.14 m in S4, rising again to 0.15 m in S5. During downward deceleration (Phase 2), displacements remain around 0.18–0.20 m, with no clear pattern associated with increasing load. On the way up (Phases 3 and 4), the displacement in Phase 3 varies subtly (between 0.18 m and 0.21 m), with S5 being the set that reaches the highest value (0.21 m), although without significant differences. In contrast, the upward deceleration phase (Phase 4) does show a significant reduction in displacement, from 0.16 m in S1 to 0.12 m in S5, with significant differences between all load conditions, grouped between the lowest load level (S1), the intermediate levels (S2 and S3) and the highest levels (S4 and S5). When evaluating the combined range of the two ascending phases (P3 + P4), stability around 0.33–0.34 m is observed, with no statistically significant findings. This indicates that, although the range of ascent is the same, the range of movement during which acceleration and deceleration of the CoM occurs varies within the ascent, which evidences the need to pay attention to the mechanical behavior of the exercise, taking into consideration the four phases detected.

Regarding *velocity*, the lowering (P1 and P2) shows negative mean and maximum values, indicating a downward displacement, with no significant differences. During P1, mean velocity decreases slightly as the load increases, presenting values between −0.50 m/s and −0.46 m/s, while P2 shows values between −0.55 m/s and −0.42 m/s. In both phases, the peak values are identical (up to −0.90 m/s at the lightest load) because the peak velocity in P2 occurs an instant after the end of P1, as a result of the segmentation methodology of the phases.

In contrast, the lifting (P3 and P4) shows positive values, indicating upward displacement. During the acceleration phase of the ascent (P3), the average velocities drop from 0.61 m/s in S1 to 0.43 m/s in S5, while the maximum velocities drop from 1.04 m/s to 0.77 m/s, indicating a progressive slowing of the ascent under conditions of increased weight. These reductions, in several cases, are statistically significant, especially when comparing the sets with light and moderate loads (S1–S2–S3) versus higher loads (S4–S5). In this sense, the proposed four-phase model enables the identification of differences in average velocity in P3 that cannot be observed when analyzing the lifting in aggregate (P3 + P4). Lastly, the final phase of the ascent (Phase 4) presents average velocity lower than those of P3, with a range that goes from 0.58 m/s (S1) to 0.46 m/s (S5). For its part, maximum velocity coincides with that detected in P3, for the same reason as explained in the description of P1 and P2.

The *acceleration* confirms this trend. In the downward acceleration phase (P1), the average and maximum values are negative, between −3.03 m/s^2^ and −2.48 m/s^2^ on average, with no statistically significant differences with increasing load. The lowering deceleration phase (Phase 2) and the upward propulsion phase (Phase 3) show positive accelerations; both show noticeable decreases in peak acceleration at higher loads, being more marked in the concentric phase (Phase 3), with values going from 5.78 m/s^2^ (S1) to 2.50 m/s^2^ (S5). This suggests a reduced ability to apply momentum to the CoM at higher load levels, reflected in the appearance of significant differences between low and moderate versus high load conditions. Finally, upward deceleration (Phase 4) again shows negative accelerations (from −3.88 m/s^2^ to −3.05 m/s^2^ on average) that do not differ significantly across load conditions, indicating a relatively constant final deceleration pattern despite increasing load.

### 3.4. Dynamics

The assessment of mechanical power and mechanical work provides information on the capacity to generate force–velocity and the energy cost in each phase of the movement.

In terms of *mechanical power*, an inverse relationship between increasing load and mechanical power levels is evident, especially in the concentric acceleration phase (Phase 3). Thus, average mechanical power decreases from approximately 3.22 W/kg in S1 to 1.91 W/kg in S5, while peak values decrease from 5.48 W/kg to 3.63 W/kg. Although the differences do not reach statistical significance, there is a clear tendency to produce lower peak mechanical power with heavier loads, consistent with the drop in velocity during ascent. On the other hand, the eccentric phases (Phase 2 and Phase 4) show negative mechanical power values, indicating the absorption of mechanical energy by the joints to produce the braking of the movement. In these phases, no marked changes linked to the increase in load are identified, although a slight decrease in the magnitude of the negative peak is observed when the resistance is higher, especially in Phase 2. It should be noted that the average mechanical power during the ‘rise’ approaches zero, as the mechanical power of phases 3 and 4 offset each other.

The *mechanical work* also follows the pattern of alternating energy production and absorption between phases. Thus, in the downward acceleration (Phase 1), positive average values between 0.63 J/kg and 0.78 J/kg are recorded, with a slight increase as the load increases. In the downward deceleration phase (Phase 2), the mechanical work is negative, and its magnitude also increases as the resistance increases, from −0.59 J/kg in S1 to −0.77 J/kg in S5. During the upward propulsion (Phase 3), the results are of positive sign but do not follow a defined pattern, and in the final deceleration phase (Phase 4), the mechanical work is negative, with no evident tendencies but highlighting, for both phases, the second set, with values of greater magnitude. When the two lifting phases are combined (P3 + P4), some significant differences are found, even though the total mechanical work remains close to zero, showing that the energy input in the concentric phase tends to be compensated by the energy absorption during the lifting deceleration.

## 4. Discussion

In scientific literature, the number of movement phases detected, and their method of delimitation, differ according to the author and the objective of the study. On the one hand, there is a traditional approach, in which exercise is segmented into two main stages: the (imprecisely called) eccentric or lowering phase and the (imprecisely called) concentric or lifting phase [61,62,63,64]. In these works, the usual criterion to identify the transition between both phases is the moment when the vertical velocity of the CoM changes sign (going from negative to positive) or, equivalently, when the subject initiates the upward pushing action [62,65]. Authors such as Wagle et al. [66] have analyzed the movement by introducing variations related to eccentric loading or the inclusion of pauses between repetitions, while maintaining the essential distinction between the downward and upward phases. This phase model does not consider the possibility of downward movement occurring without eccentric action or upward movement occurring without concentric action.

In contrast, the proposal presented in this paper provides a more detailed segmentation into four phases, establishing as a segmentation criterion the changes of sign in mechanical power (determined by the relationship between load, vertical velocity, and CoM acceleration), which enables a more precise identification of the eccentric and concentric control intervals. Thus, while the majority literature concentrates on two major phases—descent and ascent—this proposal refines the analysis into four phases, considering the variation in mechanical power. To the best of the authors’ knowledge, no previous study has described these four phases in squat exercises, although there is published evidence of this methodology in the bench press exercise [67] and, additionally, the present findings for phases 3 and 4 are almost in line with other studies that differentiate between the propulsive phase and the braking phase [68,69].

The four-phase model proposed in this paper breaks down the movement into (1) lowering acceleration, (2) lowering deceleration, (3) lifting acceleration, and (4) lifting deceleration. Phase 1 can be compared to the negative impulse that occurs during a countermovement jump that is not caused by the subject’s direct action [70,71], and phase 4 is similar to its opposite because, if the concentric action were to continue to the end of the vertical trajectory, the CoM would not decelerate and would result in a vertical projection beyond its initial position.

The phased analysis of CoM behavior during HS also yields useful information to understand the kinematic and dynamic adaptations of exercise execution as a function of load levels. Unlike most previous studies using barbell displacement as an indicator of movement [72,73,74], this work directly assesses the CoM of the combined system (subject + barbell), which constitutes a significant methodological advancement in terms of mechanical accuracy.

In relation to kinematics, the results of this study confirm a progressive reduction in CoM velocity and acceleration with increasing load, especially during the propulsion phase (Phase 3), which is consistent with patterns described in previous studies [62,75,76,77]. The decrease in peak velocity from 1.04 m/s to 0.77 m/s and peak acceleration from 5.78 m/s^2^ to 2.50 m/s^2^ between the lowest and highest loads supports the already established inverse relationship between load and execution velocity [77], observed even when considering only barbell movement. This study, however, shows that this trend also holds when considering the overall CoM, which reinforces its validity as a generalizable biomechanical phenomenon. Based on the statistically significant differences observed, it can be determined that knowing the maximum acceleration allows for the identification of more differences than analyzing velocity alone, particularly in P2. This highlights the importance of complementing the information provided by velocity with acceleration data.

In terms of displacement, a maintenance of the total range of the ascent (P3 + P4) is observed, suggesting that the subjects internally adjust the distribution of movement between phases, but not the total range. In the final deceleration of the ascent (P4), as the load is increased, a significant decrease in displacement is observed, attributable to an adjustment in the distribution of the range of movement in which each phase is developed, increasing the distance covered in P3 and decreasing, proportionally, the distance in P4. This observation partially coincides with Larsen [74], but with greater precision, by decomposing and quantifying the deceleration phase (P4) independently, and identifying how its relative contribution decreases with increasing load.

From a temporal perspective, it is noteworthy that the redistribution of the durations between phases shows that, at high loads, subjects reduce the downward acceleration time (P1) and increase the duration of the eccentric deceleration phase (P2). This strategy could be explained as a safety or efficiency mechanism, intended to minimize descent velocity at the start of the deceleration phase, thus facilitating the change in direction, but possibly impairs the adjustment to an optimal acceleration course [78], which maximizes mechanical power values. With respect to the ascent, Larsen [74] already noted the existence of a deceleration phase in the ascent (equivalent to P4), but the present work extends this finding by showing how the duration of the P3 and P4 phases is adjusted in response to the load.

The analysis of mechanical power is consistent with studies such as those of Sinclair [76] and Zink [77] in documenting a clear reduction in peak mechanical power values as load increases, both in absolute terms and relative to body weight. This pattern reinforces the hypothesis that optimal mechanical power is achieved at moderate loads, and that mechanical power production is limited under conditions of high resistance. By empirically demonstrating that, in the half squat exercise, the maximum level of concentric mechanical power (average and maximum) of the end-effector (CoM) is developed at the load condition of 35% 1RM, the theory of Hill’s [79] model is substantiated. This model states that “the greatest rate of external work performance should occur at a load equal to approximately 30% of the isometric tension”, understanding that the isometric tension will be very close (slightly greater) to that produced at 100% of the 1RM.

Regarding mechanical work, the findings of the present study enable the distinction between energy absorption and production phases (eccentric and concentric, respectively). The progressive increase in negative mechanical work during lowering deceleration (Phase 2) at high loads has not previously been documented with this precision, as previous studies do not disaggregate mechanical work by phase [72,73]. Furthermore, the tendency of the total mechanical work to approach zero during the ascent (P3 + P4), a product of the trade-off between positive and negative mechanical work, provides a novel interpretation of the energy balance of the gesture, hardly considered in previous work.

A reduction in the magnitude of various kinematic and mechanical parameters has also been identified as load intensity increases, specifically during phases 2 and 3. This trend is also consistent with Hill’s muscle mechanical model [79]. Furthermore, the observed pattern supports the proposed hypotheses and aligns with prior research that has reported an inverse relationship between load and velocity in weightlifting movements [80,81,82].

Overall, mechanical power and joint mechanical work show that progressively increasing load results in increased stress in both the eccentric and concentric phases. The hip emerges as the main biomechanical resource for supporting and elevating progressively heavier loads, especially when combining its high capacity to generate mechanical power with the need to absorb and produce mechanical work in the most demanding parts of the movement. The knee remains a fundamental pillar in force absorption and generation, but its relative contribution may be partially relegated as the demand intensifies in the heavier sets. The ankle, finally, retains a secondary role, although it adopts slight increases in participation at certain times, especially at the end of the ascent, without matching the involvement of the proximal joints.

In summary, the present study defined and applied the Power-Based Training (PBT) approach to the biomechanical evaluation of the half squat exercise, used here as a reference for lower-limb training. The results confirm trends previously reported in the literature, such as the decrease in velocity, acceleration, and mechanical power as exercise load increases. However, they provide a more accurate representation of the behavior of the subject–barbell system by integrating the combined CoM, segmenting the motor pattern into functional phases, and evaluating the pattern’s adaptation as a function of the applied load. Furthermore, the findings demonstrate that the temporal and kinematic patterns of the HS exercise should be analyzed phase by phase, rather than treating the movement as an indivisible unit (cycle or repetition) or merely dividing it into two phases based solely on movement direction.

This perspective creates new opportunities for future research that integrates joint mechanical power (JMP) and joint mechanical work (JMW) through inverse dynamics analysis; variability studies grounded in dynamic systems theory [83] and uncontrolled manifold hypothesis [84,85], synergy analysis [86,87], usability and ergonomics (efficiency, safety, and comfort) [88].

## 5. Conclusions

The results of this research highlight the relevance of analyzing the half squat (HS) from a systemic structural approach and based on mechanical power, thus defining the Power-Based Training (PBT) approach, in contrast to the traditional division into only two phases (descent and ascent). The segmentation into four phases (lowering acceleration, lowering deceleration, lifting acceleration, and lifting deceleration) makes it possible to identify more precisely the internal dynamics of each interval and the way in which the mechanical demands vary according to the external load. This analytical strategy also demonstrated the importance of assessing the CoM of the complete system (subject + bar), which provides a more accurate view than the exclusive study of the trajectory of the bar.

The progressive increase in load intensity evidenced a reduction in peak velocity and acceleration, as well as in mechanical power output, especially in the propulsion phase (phase 3). In addition, a redistribution in the duration and magnitude of displacement was observed during the total HS cycle; in particular, the upward deceleration phase (phase 4) was shortened at higher load levels, while the propulsion phase showed a prolongation and lower peak velocity. These findings corroborate the inverse relationship between load magnitude and the ability to rapidly generate force–velocity, ratifying the validity of the mechanical model proposed in the literature.

The analysis of mechanical power and mechanical work, in turn, reveals the prominent role of the eccentric phases in the absorption of mechanical energy for the braking of the movement, in the final stretch of the descent (phase 2). The fact that the sum of the positive and negative work in the ascent (phases 3 and 4) tends to equalize confirms the need to break down the movement into functional phases, since high loads do not modify the total range of displacement but do require adjustments in execution to lift the bar effectively.

The detailed biomechanical characterization of HS provides practical evidence for the planning and optimization of lower body strength training, as it helps to determine more accurately the neuromuscular demands of each phase and to guide the prescription of loads and intensities in a specific way. Likewise, it lays the groundwork for future work integrating inverse dynamic analysis, gesture variability, and muscle synergies, to improve dosage and performance control in countermovement exercises such as the half squat, as well as to evaluate the mechanical joint contribution and the symmetry or asymmetry in the execution of the exercises.

## Figures and Tables

**Figure 1 bioengineering-12-00603-f001:**
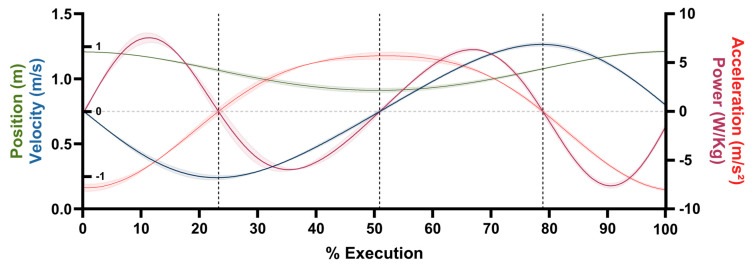
CoM movement description: position (m), velocity (m·s^−1^), acceleration (m·s^−2^), and mechanical power (W/Kg) of one interval 1 (20% 1RM) subject sequence (3 repetitions). Phases (separated by dotted lines). Lines represent the mean, and the shaded areas represent the standard deviation. Repetitions were normalized over 100% of execution within 1000 frames. Left vertical axis represents position (left ticks) and velocity (right ticks). Right vertical axis represents acceleration and mechanical power.

**Figure 2 bioengineering-12-00603-f002:**
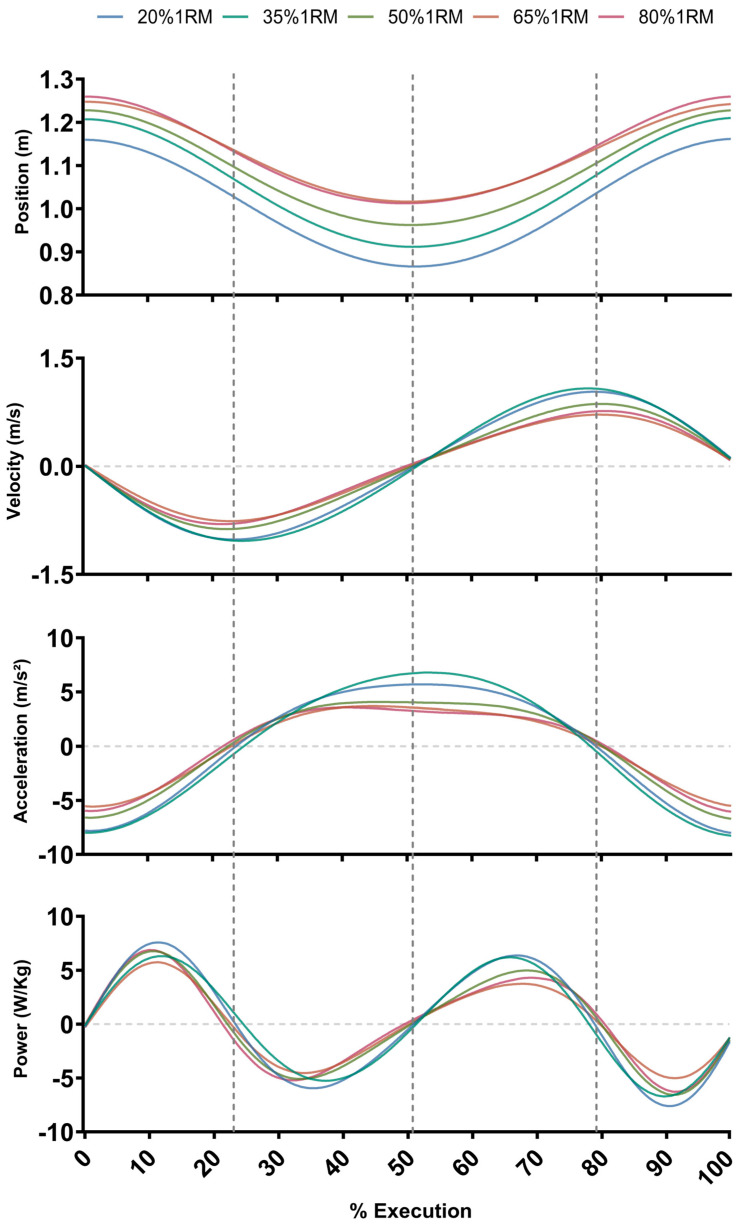
CoM movement averaged time series: position (m), velocity (m·s^−1^), acceleration (m·s^−2^), and mechanical power (W/Kg) of one subject sequence (3 repetitions) at all intervals: 1 to 5 (20% 1RM–80% 1RM). Phases appear separated by vertical dotted lines. Repetitions were normalized over 100% of execution within 1.000 frames.

**Table 1 bioengineering-12-00603-t001:** Kinematic and dynamic variables of CoM behavior associated with HS execution in each phase of the movement and for each load level. Mean and maximum values are shown, as well as the significant differences (SD) found between the values of the sets. Sets in which SD were identified are highlighted in bold, with superscripts indicating the set or sets against which the differences were found.

		Avg. Duration		Position (m)		Velocity (m/s)				Acceleration (m/s^2^)				Mechanical Power (W/Kg)				Mechanical Work (J/Kg)			
**Phase**	**Sets**	**(s)**	**(%)**	**Range**	**SD**	**Avg.**	**SD**	**Max**	**SD**	**Avg.**	**SD**	**Max**	**SD**	**Avg.**	**SD**	**Max**	**SD**	**Avg.**	**SD**	**Max**	**SD**
**P1**	S1	0.42	28.9%	0.17		−0.48		−0.90		−3.03		−5.04		2.43		4.79		0.63		0.69	
	S2	0.33	24.4%	0.16		−0.50		−0.89		−3.05		−5.20		2.80		5.54		0.71		0.79	
	S3	0.34	24.3%	0.16		−0.46		−0.80		−2.52		−4.45		2.36		4.65		0.67		0.74	
	S4	0.31	22.5%	0.14		−0.46		−0.78		−2.58		−4.47		2.67		5.21		0.75		0.80	
	S5	0.31	21.4%	0.15		−0.46		−0.77		−2.48		−4.39		2.76		5.47		0.78		0.84	
**P2**	S1	0.34	25.0%	0.19		−0.55		−0.90		3.11		**5.40**	** ^4,5^ **	−2.35		−4.10		−0.59		−0.65	
	S2	0.38	28.2%	0.20		−0.52		−0.89		2.74		**4.63**	** ^4,5^ **	−2.34		−4.20		−0.68		−0.76	
	S3	0.39	27.5%	0.18		−0.45		−0.80		2.32		**3.59**	** ^4,5^ **	−2.02		−3.61		−0.65		−0.71	
	S4	0.40	28.6%	0.18		−0.44		−0.78		2.15		**3.10**	** ^1,2,3,5^ **	−2.10		−3.58		−0.73		−0.78	
	S5	0.41	28.5%	0.18		−0.42		−0.77		1.97		**2.81**	** ^1,2,3,4^ **	−2.05		−3.82		−0.77		−0.83	
**P3**	S1	0.28	22.0%	0.18		**0.61**	** ^4,5^ **	**1.04**	** ^4^ **	**4.00**	** ^4,5^ **	**5.78**	** ^4,5^ **	3.22		5.48		0.81		0.86	
	S2	0.30	23.9%	0.19		**0.59**	** ^4,5^ **	**1.02**	** ^4,5^ **	**3.44**	** ^3,4,5^ **	**4.98**	** ^4,5^ **	3.31		5.76		0.93		0.99	
	S3	0.34	25.0%	0.18		**0.51**	** ^5^ **	0.86		**2.51**	** ^2,5^ **	**3.56**	** ^4,5^ **	2.43		4.10		0.78		0.82	
	S4	0.38	28.0%	0.19		**0.48**	** ^1,2^ **	**0.81**	** ^1,2^ **	**2.12**	** ^1,2^ **	**3.09**	** ^1,2,3,5^ **	2.23		3.69		0.82		0.84	
	S5	0.45	31.2%	0.21		**0.43**	** ^1,2,3^ **	**0.77**	** ^2^ **	**1.75**	** ^1,2,3^ **	**2.50**	** ^1,2,3,4^ **	1.91		3.63		0.81		0.84	
**P4**	S1	0.34	24.1%	**0.16**	** ^2,3,4,5^ **	0.58		**1.04**	** ^4^ **	−3.88		−6.12		−3.10		−5.83		−0.76		−0.80	
	S2	0.33	23.5%	**0.15**	** ^1,4,5^ **	0.56		**1.02**	** ^4,5^ **	−3.91		−6.18		−3.68		−7.11		−0.88		−0.95	
	S3	0.34	23.3%	**0.14**	** ^1,4, 5^ **	0.47		0.86		−3.04		−5.04		−2.88		−5.58		−0.75		−0.79	
	S4	0.29	20.9%	**0.13**	** ^1, 2, 3^ **	0.47		**0.81**	** ^1,2^ **	−3.15		−5.08		−3.21		−5.90		−0.79		−0.82	
	S5	0.27	18.8%	**0.12**	** ^1, 2, 3^ **	0.46		**0.77**	** ^2^ **	−3.05		−5.08		−3.22		−6.03		−0.79		−0.82	
**Raise**	S1	0.63	46.1%	0.34		0.59		**1.04**	** ^4^ **	**0.06**	** ^2,3,4,5^ **	**5.78**	** ^4,5^ **	0.10		5.48		**0.05**	** ^4, 5^ **	**0.05**	** ^4, 5^ **
**(P3 + P4)**	S2	0.63	47.3%	0.34		0.57		**1.02**	** ^4,5^ **	**0.00**	** ^1^ **	**4.98**	** ^4,5^ **	0.08		5.76		**0.04**	** ^4, 5^ **	**0.05**	** ^4, 5^ **
	S3	0.68	48.3%	0.33		0.49		0.86		**−0.03**	** ^1^ **	**3.56**	** ^4,5^ **	0.04		4.10		0.03		0.03	
	S4	0.67	48.9%	0.32		0.47		**0.81**	** ^1,2^ **	**−0.04**	** ^1^ **	**3.09**	** ^1,2,3,5^ **	0.04		3.69		**0.02**	** ^1, 2^ **	**0.02**	** ^1, 2, 5^ **
	S5	0.72	50.0%	0.33		0.44		**0.78**	** ^2^ **	**−0.03**	** ^1^ **	**2.50**	** ^1,2,3,4^ **	0.03		3.63		**0.02**	** ^1, 2^ **	**0.02**	** ^1, 2, 4^ **

## Data Availability

The data presented in this study are available on request from the corresponding author. The data are not publicly available due to ethical considerations.

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
