# Peer review of "Half Squat Mechanical Analysis Based on PBT Framework"

_bioengineering, 2025, doi:10.3390/bioengineering12060603_

Round 1
Reviewer 1 Report
Comments and Suggestions for Authors
This manuscript examines the half squat (HS) using a systemic-structural approach grounded in Power-Based Training (PBT). Five weightlifters from the Mexican national team (categories U17, U20, and U23) participated, performing 5 repetitions per series of HS with increasing loads (20%, 35%, 50%, 65%, and 80% of the 1RM). The findings revealed a notable decrease in velocity, acceleration, and mechanical power as the load increased, along with changes in the duration and range of movement per phase. Consequently, the reviewer believes these results underscore the necessity of a detailed analysis to comprehend the neuromuscular demands of HS and to enhance its training. The PBT approach and global CoM analysis offer a more precise understanding of exercise mechanics, aiding its application in future studies. Therefore, the reviewer recommends major revisions for this manuscript in this journal.
- As a scientific paper, having only one figure and one table is insufficient for readers to fully grasp the content. The reviewer suggests including more figures and data representations, such as the composition of PBT frameworks and visualizations of the data in Table 1.
- Compared to existing articles on magnetite-based nanocomposites, this paper does not emphasize significant differences or unique features. The authors should clarify the strengths and comparisons within the manuscript.
- Could you provide more details on the experimental design and procedure using the PBT frameworks?
- The title referring to the PBT Framework is not suitable. Please use the full name instead of the abbreviation for better clarity.
- Some language issues need attention.
Line 14-15: through which four phases of the movement were determined (acceleration and deceleration in descent, acceleration and deceleration in ascent).
6. The typography and layout of the charts are crucial for effectively conveying the content. Based on my review of the diagrams, there are areas for improvement. It is recommended to ensure consistent sizing of charts and uniform arrangement of elements like titles and notes. For instance, the text and numbers in Table 1 are not very clear and appear thickly dotted, which could be categorized differently.
Comments on the Quality of English Language- Some language issues need attention.
Line 14-15: through which four phases of the movement were determined (acceleration and deceleration in descent, acceleration and deceleration in ascent).
Author Response
Dear Reviewer,
I would like to sincerely thank you for your valuable comments and suggestions provided during the review process. The manuscript has been thoroughly revised, and the corresponding changes have been implemented. To facilitate your evaluation, all modifications have been highlighted in yellow within the article text.
Below, we respond point by point to your observations:
Comment: “Having only one figure and one table is insufficient for readers to fully grasp the content. Include more figures and data representations, such as PBT frameworks and visualizations of Table 1.”
Response: A second figure has been added, presenting each variable (position, velocity, acceleration, and mechanical power) individually and illustrating their variation across the five evaluated load levels.
Comment: “Compared to existing articles on magnetite-based nanocomposites, this paper does not emphasize significant differences or unique features.”
Response: This comment appears to have been included by mistake, as our manuscript does not address magnetite-based nanocomposites but the biomechanical analysis of the Half Squat. We kindly ask the reviewer to revisit this comment.
Comment: “Provide more details on the experimental design and procedure using the PBT frameworks.”
Response: Acknowledging the potential confusion caused by the description of the experimental design and procedure, the first paragraph of Section 2.4.3, “Determination of the phases,” has been improved for clarity.
Comment: “The title referring to the PBT Framework is not suitable. Please use the full name instead of the abbreviation.”
Response: A review of recent scientific literature shows that the use of abbreviations in article titles—especially those referring to methodological approaches or paradigms—is common practice. Moreover, in this case, Power-Based Training is established as a training method brand. For these reasons, we believe maintaining the acronym in the title helps reinforce its recognition among the scientific and professional community. Nevertheless, we defer to the editor’s final decision on this matter.
Comment: “Some language issues need attention. Line 14-15: through which four phases of the movement were determined (acceleration and deceleration in descent, acceleration and deceleration in ascent).”
Response: The terminology used in the specified lines has been corrected, and the language throughout the manuscript has been revised for clarity and precision.
Comment: “The typography and layout of the charts are crucial for effectively conveying the content. Based on my review of the diagrams, there are areas for improvement. It is recommended to ensure consistent sizing of charts and uniform arrangement of elements like titles and notes. For instance, the text and numbers in Table 1 are not very clear and appear thickly dotted, which could be categorized differently.”
Response: The design of the figures has been improved, and visual consistency across all figures and tables in the manuscript has been carefully verified.
I remain at your disposal for any further suggestions that may help enhance the quality of the manuscript.
Sincerely,
Miguel Rodal
Reviewer 2 Report
Comments and Suggestions for Authors
Dear Authors
The interesting manuscript “Half Squat Mechanical Analysis based on PBT Framework” clear and relevant for the Biomechanics field. This study written in an appropriate way. The results provide an advancement of the current knowledge.
The figure diagram [Figure 1. CoM movement description: position (m), velocity (m·s−1), acceleration .... ] presented very appropriately ]. Presented findings partly evidence the importance of a this type detailed analysis to understand the neuromuscular
demands and facilitating its application in future planning and monitoring effects.
With respects, Reviewer
Author Response
Dear Reviewer,
I would like to thank you for the time dedicated to reviewing the manuscript and for your kind comments, especially your positive observation regarding the presentation of Figure 1.
Your feedback reinforces our conviction about the relevance of the approach adopted and its potential positive impact on the planning and monitoring of strength training.
I remain open to any further suggestions you may have that could help enhance the quality of the work.
Sincerely,
Miguel Rodal
Round 2
Reviewer 1 Report
Comments and Suggestions for Authors
None, the paper can be accepted after this revision.
Author Response
Dear Reviewer,
I would like to thank you for the time dedicated to reviewing the manuscript and for your comments and suggestions.
Sincerely,
Miguel Rodal